# Factors associated with venous collaterals in patients with cerebral venous thrombosis

**Phirat Petchprom[1], Nuttha Sanghan[2], Rujimas Khumthong[2], Suwanna Setthawatcharawanich[1], Pornchai Sathirapanya[1], Rattana Leelawattana[3], Pat Korathanakhun[1] ***

**1** Division of Neurology, Department of Internal Medicine, Faculty of Medicine, Prince of Songkla University, Hatyai, Songkhla, Thailand, **2** Division of Neuroradiology, Department of Radiology, Faculty of Medicine, Prince of Songkla University, Hatyai, Songkhla, Thailand, **3** Division of Endocrinology, Department of Internal Medicine, Faculty of Medicine, Prince of Songkla University, Hatyai, Songkhla, Thailand

* patosk120@gmail.com

## Abstract

### Objectives

To identify the factors associated with venous collaterals in Thai patients with cerebral venous thrombosis.

### Materials and methods

This retrospective 20-year cohort study enrolled patients diagnosed with cerebral venous thrombosis between January 2002 and December 2022. Data was collected from the electronic medical record, and venous collaterals were independently reviewed by two neuroradiologists using the Qureshi classification. Patients with and without venous collaterals were compared. Significant factors (P<0.05) in the univariate analysis were recruited into the multivariate logistic regression analysis to determine independently associated factors.

### Results

Among 79 patients with cerebral venous thrombosis, the prevalence of venous collaterals at the initial neuroimaging was 25.3%. In the univariate analysis, patients with cerebral venous thrombosis and venous collaterals were significantly younger (37.0±13.9 years vs. 44.9 ±17.4 years, P = 0.048), more often had occlusion in the superior sagittal sinus (80.0% vs. 54.2%, P = 0.041), and were associated with hormonal exposure (35.0% vs. 6.8%, P = 0.002). Multivariate logistic regression analysis revealed occlusion in the superior sagittal sinus (adjusted odds ratio [aOR] 3.581; 95% confidence interval [95% CI] 1.941–13.626; P = 0.044) and hormonal exposure (aOR 7.276, 95% CI 1.606–32.966, P = 0.010) as independent factors associated with venous collaterals in cerebral venous thrombosis.

### Conclusions

In this cohort, the prevalence of venous collaterals was 25.3%. Occlusion in the superior sagittal sinus and hormonal exposure were independently associated with venous collaterals in patients with cerebral venous thrombosis.

**Data Availability Statement:** All relevant data are within the manuscript and its Supporting Information files.

**Funding:** Only the author PK received a funding for this research from the Faculty of Medicine, Prince of Songkla University (funding number: REC.66-122-14-1). The funder did not involve in the study design, data collection and analysis, decision to publish, or preparation of the manuscript. The URL of the funder is https://medinfo.psu.ac.th/home/index.php.

**Competing interests:** The authors have declared that no competing interests exist.

## Introduction

Cerebral venous thrombosis (CVT) is a potentially life-threatening veno-occlusive disease. Occlusion of intracranial venous drainage could lead to permanent brain damage. With an acute thrombus in the venous channel, the venous pressure will increase suddenly, causing venous congestion. The effect of venous congestion varies based on the occlusive site. In small venous branches, like the cortical vein, occlusion will affect a focal brain area and result in venous infarction. If the thrombus is propagated through the main venous system and occludes a single main venous structure (namely the superior sagittal sinus) or a bilateral paired structure occlusion (specifically, the bilateral transverse or sigmoid sinus), the serious conditions including increased intracranial pressure and intracranial hemorrhage may occur.

Venous collaterals arising from occluded sites are occasionally found in the radiographic studies of patients with CVT. However, it's significance is not well established. Based on the current evidence, only 4 studies had objectively investigated the venous collateral in the different aspects. In terms of the clinical outcome prediction, these studies showed discordant results; either positive, negative or undetermined effect on the clinical outcome [1–3]. In the viewpoint of radiological outcome, a single study revealed the prediction of early recanalization [4].

Apart from the predictive role of venous collateral, several issues have never been mentioned. These gaps of knowledge include the factor contributing to venous collateral generation, the proper neuroimaging technic to evaluate venous collateral, the reliable classification for venous collateral grading and the correlation of the classification to either the clinical outcome and the complications. In this study, we aimed to identify the independent factors associated with venous collaterals at the diagnosis of CVT. In order to correct the drawback from the previous study; in regard of the non-uniform venous collateral grading, we apply the Qureshi classification in our study.

## Methods

### Patients and methods

This was a retrospective 20-year cohort study conducted in a major referral hospital covering 14 provinces in southern Thailand. The inclusion criteria were as follows: 1) 18 years old or older, 2) admitted in the inpatient department, and 3) primary neuroimaging available for review (contrast-enhanced magnetic resonance imaging [MRI] of the brain, magnetic resonance venography [MRV] of the brain, or cerebral angiography). Although the CT venography of the brain is considered as a reliable method for CVT diagnosis, the visualization of venous collateral is problematic owing to the limited resolution of the CT venography. Therefore, the CT venography was not set in the inclusion criteria.

Inpatient records of the patients diagnosed with spontaneous CVT between January 2002 and December 2022 were screened using the International Classification of Diseases, 10th revision (ICD-10), using the following codes: G08 (intracranial and intraspinal phlebitis and thrombophlebitis), I636 (cerebral infarction due to CVT), I676 (non-pyogenic thrombosis of the intracranial venous system), and O225 (CVT in pregnancy). Cases with wrong diagnoses, incomplete medical records, or lacking primary neuroimaging data were excluded. Patients with cavernous sinus thrombosis were also excluded.

### Data collection

Baseline characteristics, clinical presentations, initial functional score, neuroimaging characteristics, treatment, and clinical outcome were collected from the electronic medical record. Venous collaterals were independently reviewed by two neuroradiologists. The interrater reliability of

venous collateral grading was calculated by using percentage agreement. Any discordant radiographic interpretation was discussed until a consensus was reached. The relevant data were accessed for research purposes by the authorized research team members between May 1st, 2023 and June 30th, 2023. The personal identity data of the individual participants in the research was protected and cannot be accessed according to the Personal Data Protection Act (2019).

### Definition of terms

To unify ambiguous terms that are often used in the literature, we objectively classified clinical presentations into four distinctive neurological syndromes as follows: 1) seizure (any type of seizure according to the International League Against Epilepsy (ILAE) 2017 Classification [5]) 2) focal neurological deficits (any abnormal focal neurological deficits), 3) isolated increased intracranial pressure (the collective clinical symptoms of headache, blurred vision, with or without papilledema and objective evidence of increased intracranial pressure proven by the CSF pressure over 20 cm H2O without any focal neurological deficits), and. 4)isolated headache (headache without abnormal neurologic signs and the CSF pressure under 20 cm H2O). Venous collaterals were classified according to the Qureshi classification [6] as follows: in grade 1, the collateral bypassed the occluded segment and connected it within the same dural sinus; in grade 2, the collateral bypassed the occluded segment and connected it to a different dural sinus in the same circulatory pathway; and in grade 3, the collateral bypassed the occluded segment and connected it to a different sinus in a different circulatory pathway. If multiple grades were identified, the highest grade was selected (Fig 1).

The modified Rankin scale (mRS) was used to evaluate the functional status upon initial admission and discharge. In addition, the hospital clinical outcome was categorized into non-dependency (mRS 0–2) and dependency or death (mRS 3–6). Patients who died of an intracranial condition, especially brain herniation, were marked as CVT-related deaths; all other causes were noted as non-CVT-related.

### Data analysis

Discrete data are described as numbers and percentages. Continuous data are presented as either mean ± standard deviation (SD) or median and interquartile range, depending on normality. The data were compared between patients with and without venous collaterals using the univariate chi-squared test, Mann–Whitney U test, or independent t-test. Factors with P-values <0.05 in the univariate analysis were recruited into the multivariate logistic regression analysis. Factors with P-values <0.05 in the multivariate logistic regression were determined as independently associated factors. This research study was approved by the Ethics Committee of the Faculty of Medicine, Prince of Songkla University.

## Results

After screening with the ICD-10 codes G08, I636, I676, and O225, 228 patients with CVT were identified; 149 patients were excluded due to a wrong diagnosis or a lack of available primary neuroimaging. Finally, 79 patients with CVT were enrolled in the study: 20 with venous collaterals and 59 without. The prevalence of venous collaterals at the initial neuroimaging was 25.3%.

### Comparison of baseline characteristics, clinical presentations, and radiographic findings

Patients with venous collaterals were significantly younger than those without (37.0±13.9 years vs. 44.9±17.4 years, t = -2.041, P = 0.048). There was no significant difference in terms of sex

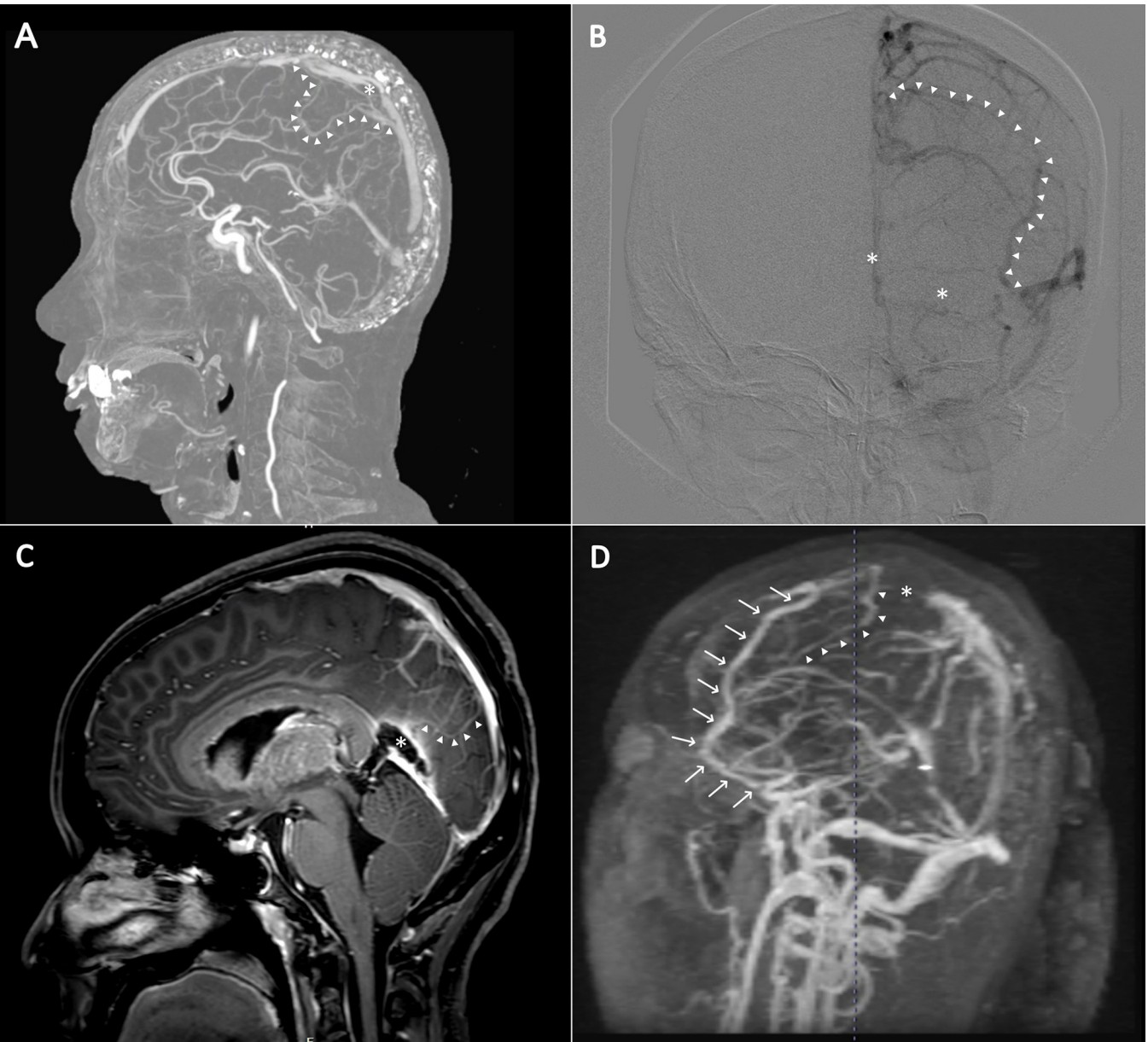

**Fig 1. Brain imaging showing venous collaterals according to the Qureshi classification.** (**A**) Venous collateral grade 1. Sagittal magnetic resonance venography (MRV) of the brain demonstrated thrombosis of the superior sagittal sinus (asterisk) and the venous collateral bypassing the occlusion site and connecting with the distal part of the same sinus (arrowhead). (**B**) Venous collateral grade 2. Coronal cerebral angiography revealed extensive thrombosis from the distal superior sagittal sinus to the left proximal transverse sinus (asterisk) and the venous collateral connecting the middle superior sagittal sinus with the transverse-sigmoid sinus junction (arrowhead). (**C**) Venous collateral grade 3. Gadolinium-enhanced sagittal magnetic resonance imaging of the brain demonstrated thrombosis of the great vein of Galen and straight sinus (asterisk) and venous collateral connecting the inferior sagittal sinus with the distal superior sagittal sinus (arrowhead). (**D**) MRV of the brain revealed thrombosis of the superior sagittal sinus (asterisk) and two venous collaterals as follows: (1) venous collateral grade 2, connecting the proximal superior sagittal sinus connecting with the Sylvian vein (arrow) and (2) venous collateral grade 3, connecting the proximal superior sagittal sinus with the inferior sagittal sinus. The patient with multiple venous collaterals was graded according to the highest grade of the Qureshi classification.

(with and without collaterals: 60.0% female vs. 64.4% female, respectively, P = 0.724) or smoking exposure (5.0% vs. 5.1%, P = 0.976) between groups.

No significant differences were found between groups in terms of clinical presentations (with and without collaterals: focal neurological deficit, 60.0% vs. 37.3%, P = 0.076; isolated

increased intracranial pressure, 30.0% vs. 17.0%, P = 0.209; isolated headache, 30.0% vs. 42.4%, P = 0.327; seizure, 45.0% vs. 40.7%, P = 0.735). The clinical severity at admission was also comparable between groups (with and without collaterals: mRS 3–5, 45.0% vs. 37.3%, P = 0.542).

To confirm the CVT diagnosis and identify venous collaterals, at least one gold-standard neuroimaging method (either brain MRI or MRV) was performed in all patients. The number of venous thrombosis sites was quantified to evaluate the clot burden. However, there was no significant difference between groups in terms of patients with more than three thrombosis sites (with and without collaterals: 50.0% vs. 30.5%, P = 0.115). Patients with venous collaterals had a significantly higher prevalence of CVT in the superior sagittal sinus than those without venous collaterals (80.0% vs. 54.2%, P = 0.041). The other CVT sites were comparable between groups (with and without collaterals: superficial cortical vein, 30.0% vs. 27.1%, P = 0.804; transverse sinus, 60.0% vs. 49.2%, P = 0.401; deep cerebral vein, 5.0% vs. 5.1%, P = 0.988; sigmoid sinus, 40.0% vs. 45.8%, P = 0.654; straight sinus, 10.0% vs. 5.1%, P = 0.435; and jugular vein, 25.0% vs. 20.3%, P = 0.661). The prevalence of intracranial hemorrhage was also similar between groups (50.0% vs. 35.6%, P = 0.254) (Table 1).

### Characteristics of venous collaterals

The venous collateral grading was independently assessed by 2 neuroradiologists. The interrater reliability calculated by percentage agreement was 84.8%. The 12 discordant venous collateral findings were concluded by consensus. In this cohort, the venous collaterals were found in 20 patients (25.3%) according to the Qureshi classification [6]. Grade-3 collaterals were the most common (11 patients), followed by grades 1 (5 patients) and 2 (4 patients).

### Comparison of etiology, treatment, and hospital outcome

Hormonal exposure (35.0%) was the most common cause of CVT in patients with venous collaterals, while idiopathic CVT (32.2%) was the most common in patients without venous collaterals. Patients with venous collaterals had a significantly higher prevalence of hormonal exposure than those without (35.0% vs. 6.8%, P = 0.002). However, no significant differences were found between groups in terms of infection (with and without collaterals: 5.0% vs. 6.8%, P = 0.778), cancer (10.0% vs. 25.4%, P = 0.147), pregnancy and puerperium (10.0% vs. 1.7%, P = 0.093), hypercoagulable state (20.0% vs. 17.0%, P = 0.757), miscellaneous etiologies (10.0% vs. 27.1%, P = 0.115), and idiopathic CVT (30.0% vs. 32.2%, P = 0.855).

Most patients in both groups underwent anticoagulant therapy (either unfractionated or low-molecular-weight heparin). The numbers of patients treated with anticoagulants (with and without collaterals: 95.0% vs. 89.8%, P = 0.482) and decompressive surgery (with and without collaterals: 10.0% vs. 6.8%, P = 0.638) were comparable between groups. There was no difference in poor outcome (mRS 3–6) at discharge between groups (with and without collaterals: 60.0% vs. 71.2%, P = 0.353). Of the 79 patients in this cohort, The overall mortality in the study cohort was 6.3%. Furthermore, the numbers of CVT-related deaths were similar between groups (100.0% vs. 75.0%, P = 0.439) (Table 2).

### Factors associated with venous collaterals in patients with CVT

Significant factors in the univariate analysis included age (with vs. without venous collaterals: 37.0±13.9 years old vs. 44.9±17.4 years old, P = 0.048), occlusion in the superior sagittal sinus (80.0% vs. 54.2%, P = 0.041), and hormonal exposure (35.0% vs. 6.8%, P = 0.002). After including these factors in the final multivariate logistic regression analysis model, hormonal exposure (adjusted odds ratio [aOR] 7.276, 95% confidence interval [CI] 1.606–32.966, P = 0.010) and

**Table 1. Comparison of the baseline characteristics and radiographic findings between patients with cerebral venous thrombosis with and without venous collaterals.**

| Factors | With venous collateral (n = 20) | | Without venous collateral (n = 59) | | $\chi^2$ | P-value |
|---|---|---|---|---|---|---|
| | n | Percentage (%) | n | Percentage (%) | | |
| Age (Mean ± standard deviation) (years) | 37 ± 13.9 | | 44.86 ± 17.4 | | t = -2.041 | 0.048 |
| Female | 12 | 60.0 | 38 | 64.4 | 0.125 | 0.724 |
| Smoking | 1 | 5.0 | 3 | 5.1 | 0.001 | 0.976 |
| Clinical presentations | | | | | | |
| Focal neurological deficit | 12 | 60.0 | 22 | 37.3 | 3.143 | 0.076 |
| Isolated increase in intracranial pressure | 6 | 30.0 | 10 | 17.0 | 1.579 | 0.209 |
| Isolated headache | 6 | 30.0 | 25 | 42.4 | 0.959 | 0.327 |
| Seizure | 9 | 45.0 | 24 | 40.7 | 0.115 | 0.735 |
| GCS 3–13 | 2 | 10.0 | 1 | 1.7 | 2.052 | 0.152 |
| mRS at admission | | | | | | |
| 0 | 4 | 20.0 | 22 | 37.3 | 2.626 | 0.757 |
| 1 | 6 | 30.0 | 12 | 20.3 | | |
| 2 | 1 | 5.0 | 3 | 5.1 | | |
| 3 | 2 | 10.0 | 7 | 11.9 | | |
| 4 | 1 | 5.0 | 3 | 5.1 | | |
| 5 | 6 | 30.0 | 12 | 20.3 | | |
| mRS group at admission | | | | | | |
| mRS 3–5 | 9 | 45.0 | 22 | 37.3 | 0.373 | 0.542 |
| mRS 0–2 | 11 | 55.0 | 37 | 62.7 | | |
| Radiographic findings | | | | | | |
| MRI | 13 | 65.0 | 42 | 71.2 | 0.270 | 0.603 |
| MRV | 11 | 55.0 | 24 | 40.7 | 1.242 | 0.265 |
| Angiography | 2 | 10.0 | 6 | 10.2 | 0.000 | 0.983 |
| Number of CVT sites | | | | | | |
| ≥ 3 sites | 10 | 50.0 | 18 | 30.5 | 2.480 | 0.115 |
| ≤ 2 sites | 10 | 50.0 | 41 | 69.5 | | |
| Site of CVT | | | | | | |
| Superficial cortical vein | 6 | 30.0 | 16 | 27.1 | 0.062 | 0.804 |
| Superior sagittal sinus | 16 | 80.0 | 32 | 54.2 | 4.158 | 0.041 |
| Transverse sinus | 12 | 60.0 | 29 | 49.2 | 0.704 | 0.401 |
| Deep cortical vein | 1 | 5.0 | 3 | 5.1 | 0.000 | 0.988 |
| Sigmoid sinus | 8 | 40.0 | 27 | 45.8 | 0.201 | 0.654 |
| Straight sinus | 2 | 10.0 | 3 | 5.1 | 0.609 | 0.435 |
| Jugular vein | 5 | 25.0 | 12 | 20.3 | 0.142 | 0.661 |
| Intracranial hemorrhage | 10 | 50.0 | 21 | 35.6 | 1.300 | 0.254 |

GCS, Glasgow Coma Scale; mRS, modified Rankin score; CVT, cerebral venous thrombosis; MRI, magnetic resonance imaging; MRV, magnetic resonance venography

occlusion in the superior sagittal sinus (aOR 3.581, 95% CI 1.941–13.626, P = 0.044) were identified as independently associated factors for venous collaterals in CVT (Table 3).

## Discussion

The role of venous collaterals in CVT has rarely been mentioned in previous studies. In our study, the prevalence of venous collaterals among patients with CVT was 25.3% according to the Qureshi classification. Superior sagittal sinus involvement and hormonal therapy were independently associated with venous collaterals.

**Table 2. Comparison of the etiology, treatment, and outcome between patients with cerebral venous thrombosis with and without venous collaterals.**

| Factors | With venous collateral (n = 20) | | Without venous collateral (n = 59) | | $\chi^2$ | P-value |
|---|---|---|---|---|---|---|
| | n | Percentage (%) | n | Percentage (%) | | |
| Etiology | | | | | | |
| Infection | 1 | 5.0 | 4 | 6.8 | 0.080 | 0.778 |
| Cancer | 2 | 10.0 | 15 | 25.4 | 2.104 | 0.147 |
| Pregnancy and puerperium | 2 | 10.0 | 1 | 1.7 | 2.820 | 0.093 |
| Hypercoagulable state | 4 | 20.0 | 10 | 17.0 | 0.095 | 0.757 |
| Hormonal exposure | 7 | 35.0 | 4 | 6.8 | 9.929 | 0.002 |
| Miscellaneous | 2 | 10.0 | 16 | 27.1 | 2.488 | 0.115 |
| Idiopathic | 6 | 30.0 | 19 | 32.2 | 0.034 | 0.855 |
| Treatment | | | | | | |
| Anticoagulant treatment | 19 | 95.0 | 53 | 89.8 | 0.494 | 0.482 |
| Decompressive surgery | 2 | 10.0 | 4 | 6.8 | 0.221 | 0.638 |
| Hospital outcome | | | | | | |
| mRS at discharge | | | | | | |
| 0 | 8 | 40.0 | 24 | 40.7 | 2.096 | 0.911 |
| 1 | 3 | 15.0 | 12 | 20.3 | | |
| 2 | 1 | 5.0 | 6 | 10.2 | | |
| 3 | 2 | 10.0 | 4 | 6.8 | | |
| 4 | 1 | 5.0 | 3 | 5.1 | | |
| 5 | 4 | 20.0 | 6 | 10.2 | | |
| 6 | 1 | 5.0 | 4 | 6.8 | | |
| CVT-related death | 1/1 | 100.0 | 3/4 | 75.0 | 0.600 | 0.439 |
| mRS group at discharge | | | | | | |
| mRS 0–2 | 8 | 40.0 | 17 | 28.8 | 0.864 | 0.353 |
| mRS 3–6 | 12 | 60.0 | 42 | 71.2 | | |

CVT, cerebral venous thrombosis; mRS, modified Rankin score

## Prevalence and characteristics of venous collaterals and limitations of collateral evaluation

As venous collaterals were not assessed in previous studies, the available data regarding their prevalence in patients with CVT is limited. Our study revealed the prevalence of venous collaterals in this population to be 25.3%, which was low compared to previous reports ranging between 66% and 89% [1, 3, 4].

Among these studies, different methods were used to quantify and classify the existence of venous collaterals, which led to difficulty in data interpretation. Sheth et al. [1] used the venous

**Table 3. Multivariate analysis of factors associated with venous collaterals in patients with cerebral venous thrombosis.**

| Factors | With venous collateral l (n = 20) | | Without venous collateral (n = 59) | | $\chi^2$ | p-value | aOR | 95% CI | P-value |
|---|---|---|---|---|---|---|---|---|---|
| | n | Percentage (%) | n | Percentage (%) | | | | | |
| Age (mean ± standard deviation) (years) | | 37.0 ± 13.9 | | 44.9 ± 17.4 | t = -2.041 | 0.048 | 1.014 | 0.977–1.053 | 0.453 |
| Superior sagittal sinus occlusion | 16 | 80.0 | 32 | 54.2 | 4.158 | 0.041 | 3.581 | 1.941–13.626 | 0.044* |
| Hormonal exposure | 7 | 35.0 | 4 | 6.8 | 9.929 | 0.002 | 7.276 | 1.606–32.966 | 0.010* |

CVT, cerebral venous thrombosis; aOR, adjusted odds ratio; CI, confidence interval

collateral scale (VCS) to grade patients in an American population in which cortical venous draining without anastomosis to a patent sinus was graded as VCS 1, while our study and Barboza et al. [3] used the Qureshi classification, which does not consider non-anastomotic veins. Therefore, the prevalence of venous collaterals might have been overestimated by Sheth et al. [1]. In addition, Zhao et al. [4] categorized venous collaterals into one of two groups based on the number of well-visualized collateral pathways: poor collaterals (0–1 well-visualized pathway) and good collaterals (2 or more well-visualized pathways). The number of patients with poor collaterals might have been overestimated by Zhao et al. [4], as patients without any visualized venous collaterals were included in this group. Barboza et al. [3] used the Qureshi classification, but we determined the prevalence of venous collaterals to be one-third of that reported by them.

Although Qureshi grade 3 was predominant in our study and that by Barboza et al. [3], we found almost eight times as many Qureshi grade 1 collaterals. However, caution is recommended when interpreting these findings due to the small sample sizes in both studies.

Some limitations of the Qureshi classification should be mentioned. Although the Qureshi classification is practical for acute or subacute CVT, using this classification in chronic CVT is problematic. The presence of multiple venous collaterals in chronic CVT generates a chaotic venous network in MRI and MRV. Therefore, they might be difficult to accurately grade, and some individuals with chronic CVT might have features of all three grades. To solve this problem, we propose defining multiple venous networks as grade 4 in the Qureshi classification, which should be validated in future research. Another issue is that the evaluation of venous collaterals by static neuroimaging methods (MRI and MRV) does not demonstrate the venous flow direction, affirming the collateral drainage. Recently, novel MRI techniques have been introduced to assess cerebral venous hemodynamics. Venous blood flow, velocity, and pulsatility can be systematically measured by several techniques, including phase-contrast MRI at 7 T and magnetic resonance velocity mapping. However, these parameters have not been validated for CVT [7]. In a recent study, Dempfle et al. [8] quantified the venous volume using Statistical Parametric Mapping 8 and Advanced Normalization Tools software in susceptibility-weighted imaging. Although the measurements produced by this method were reproducible and reliable, their clinical implications should be studied.

Hemodynamic assessment of the cerebral venous system by transcranial color-coded duplex sonography (TCCS) could demonstrate venous flow [9]. TCCS identified four distinctive pathologic patterns of venous drainage in the setting of CVT: 1) drainage diversion to the cavernous sinus and the deep cerebral vein, 2) drainage reversal in the basal veins, 3) compensatory flow to the transverse sinus, and 4) flow reversal in the transverse sinus [2]. Although normalization of the initially pathologic venous TCCS was a predictor of a favorable outcome, the clinical implications of this parameter need to be studied further considering the small sample size and the use of an operator-dependent tool.

## Association between the superior sagittal sinus and venous collaterals

Our study is the first to identify an association between the superior sagittal sinus and venous collaterals in CVT. This association could be explained in terms of anatomy. Three parts of the superior sagittal sinus receive venous drainage from different structures. The anterior third, in which the border is the coronal suture, receives venous drainage from the central veins of the prefrontal medial aspect of the brain. The middle third of the superior sagittal sinus is the main drainage site of the superficial cortical vein from the dorsolateral prefrontal region of the brain, in combination with drainage from the anterior third. The posterior third receives venous blood from the straight sinus [10]. As the superior sagittal sinus is the longest venous

channel to receive drainage from most of the superficial cortical veins, occlusion of the superior sagittal sinus directly increases the cerebral venous volume and pressure in the superficial cortical veins. Apart from the superior sagittal sinus and the Trolard vein, the superficial cortical vein also drains into two other interconnecting channels: the inferior cerebral vein (Labbe vein) to the transverse sinus and the superficial Sylvian vein to the cavernous sinus [10]. Previous studies of venous collaterals focused mainly on visualizing these interconnecting drainages [2, 4].

Apart from the classic major venous channel, other venous collaterals have occasionally been reported. In cases of a meningioma invading the superior sagittal sinus, the diploic veins, which connect the inner layer with the outer layer of the cortical bone, were illustrated as another venous collateral system in cerebral venous occlusion [11]. These diploic veins could be anatomically normal structures or compensatory venous collaterals. This question was indirectly answered via neurosurgery. The subsequent brain edema and venous congestion in patients who had the diploic veins removed during craniotomy emphasized their venous collateral role. Diploic veins are classified into four groups according to their anatomic course through the skull: 1) frontal diploic veins, which mainly drain to the anterior temporal vein; 2) anterior temporal diploic veins, which drain to the middle meningeal vein, deep temporal vein, cavernous sinus, and sphenoparietal sinus; 3) posterior temporal diploic veins, which drain to either the mastoid emissary vein or transverse sinus; and 4) occipital diploic veins, which drain to either the occipital vein or transverse sinus [12].

Besides the diploic veins, the falcine venous plexus may play a role in venous drainage. The falcine sinus is formed and functions in the fetal period. The venous drainage function of the falcine sinus gradually subsides in childhood and adulthood but remains as a remnant structure. However, recent studies in healthy adult controls using either electron microscopy or MRI discovered the function of the falcine venous plexus, which connects the superior sagittal sinus with the deep venous system. The falcine venous plexus of the posterior third of the superior sagittal sinus is a consistently dense network communicating with the inferior sagittal sinus, while those of the anterior and middle thirds are variable [13, 14]. The clinical significance of the falcine and parafalcine venous plexi has been demonstrated in superior sagittal sinus-invading meningiomas [15]. According to the anatomy of venous channels, we hypothesized that some venous structures may not be visualized under normal conditions, as the main drainage pathway could function normally. However, the venous collateral can be visualized when it starts to drain venous congestion. The superior sagittal sinus is the longest venous channel and is therefore the path of least resistance compared to a more complicated interconnecting pathway, which may support the venous collateral.

## Association between hormonal exposure and venous collaterals

Our study is the first to demonstrate an association between hormonal exposure and venous collaterals in CVT. The major source of hormonal exposure in patients with CVT is oral contraception, which mainly contains estrogen and its derivatives. The association between estrogen exposure and venous thrombotic risk has been demonstrated in clinical studies and molecular research [16]. Besides venous thromboembolism, estrogen also affects angiogenesis by mediating several chemokines, including nitric oxide and vascular endothelial growth factor, to stimulate vascular endothelial cell proliferation and migration [17]. Although the relationship between estrogen and angiogenesis was studied mainly with reference to arterial vasculature [18], the molecular mechanism of estrogen-directed angiogenesis via estrogen receptors (ERs) expressed on the vascular endothelium should be similar between arteries and veins [19].

Currently, three ER isoforms have been identified: ERα, ERβ, and G protein-coupled ER (GPER). There are no direct studies of ERs in CVT; however, a study on chronic venous disease confirmed significant increases in ERα, ERβ, and GPER expression in the venous endothelium according to disease severity [20]. At the cellular level, ER activation reduces calcium-dependent venous contraction. In addition, ER agonists facilitate venous relaxation [21]. We postulate that exposure to estrogen activates these two effects, leading to enhanced distensibility in veins, and that estrogen may play a role in venous collaterals in CVT.

### Strengths, limitations, and suggestions

The heterogeneous clinical symptoms and signs of CVT in previous studies are major drawbacks in comparing and interpreting results. We addressed this problem by categorizing the clinical presentations into clearly defined clinical syndromes. Another strength of this study was the use of reliable radiological data, which were independently validated by two neuroradiologists with the good interrater reliability. However, there are some limitations. First, the neuroimaging technic in this cohort was not uniform which might provide discrepant findings between the different method. Second, the radiological data in this study were derived mainly from either contrast-enhanced MRI or MRV of the brain, which are static neuroimaging methods and cannot demonstrate the direction of collateral flow. Assessing cerebral venous flow diversion via dynamic neuroimaging methods (like cerebral angiography, advanced MRI techniques, or TCCS) may be an area of research in the future. Another issue was the retrospective nature of this study, as some data was not retrievable; however, we tried to control for this by selecting only patients with complete data, resulting in a small sample size.

Our study had expanded the limited current knowledge of venous collaterals in CVT. We suggested the clinicians and neuroradiologists to meticulously searching for venous collaterals in particularly the CVT patient who had superior sagittal sinus thrombosis or hormonal exposure. However, our experience found a difficulty to identify venous collateral accurately from the CT venography due to it's limited image resolution. We suggested to use contrast-enhanced MRI or MRV as they provide the better visualization of venous collateral. Future research should focus on validating the Qureshi classification of venous collaterals in terms of clinical outcome, the association with secondary intracranial hemorrhage, and the correlation with recanalization.

### Conclusion

In the present study, 25.3% of patients with CVT had venous collaterals. Hormonal exposure and occlusion of the superior sagittal sinus were identified as independent factors associated with venous collaterals in CVT. These two factors may have clinical implications in guiding clinicians and neuroradiologists to meticulously evaluate venous collaterals in neuroimaging.

### Supporting information

**S1 File. Raw data of CVT cohort.**
(XLSX)

### Acknowledgments

Authors would like to thanks Mr. Jarernporn Kawla-ierd for assisting in the data analysis.

## Author Contributions

**Conceptualization:** Phirat Petchprom, Nuttha Sanghan, Suwanna Setthawatcharawanich, Pornchai Sathirapanya, Rattana Leelawattana, Pat Korathanakhun.

**Data curation:** Phirat Petchprom, Nuttha Sanghan, Rujimas Khumthong, Pat Korathanakhun.

**Formal analysis:** Phirat Petchprom, Nuttha Sanghan, Rujimas Khumthong, Pat Korathanakhun.

**Funding acquisition:** Pat Korathanakhun.

**Investigation:** Nuttha Sanghan, Rujimas Khumthong, Pat Korathanakhun.

**Methodology:** Phirat Petchprom, Nuttha Sanghan, Rujimas Khumthong, Pornchai Sathirapanya, Pat Korathanakhun.

**Project administration:** Nuttha Sanghan, Rujimas Khumthong, Suwanna Setthawatcharawanich, Pornchai Sathirapanya, Rattana Leelawattana, Pat Korathanakhun.

**Resources:** Suwanna Setthawatcharawanich, Pornchai Sathirapanya, Pat Korathanakhun.

**Software:** Suwanna Setthawatcharawanich.

**Supervision:** Nuttha Sanghan, Rujimas Khumthong, Suwanna Setthawatcharawanich, Pornchai Sathirapanya, Rattana Leelawattana, Pat Korathanakhun.

**Validation:** Suwanna Setthawatcharawanich, Pornchai Sathirapanya, Rattana Leelawattana.

**Visualization:** Nuttha Sanghan, Rujimas Khumthong, Suwanna Setthawatcharawanich, Pornchai Sathirapanya, Rattana Leelawattana, Pat Korathanakhun.

**Writing – original draft:** Phirat Petchprom, Nuttha Sanghan, Rujimas Khumthong, Suwanna Setthawatcharawanich, Pornchai Sathirapanya, Rattana Leelawattana, Pat Korathanakhun.

**Writing – review & editing:** Phirat Petchprom, Nuttha Sanghan, Rujimas Khumthong, Suwanna Setthawatcharawanich, Pornchai Sathirapanya, Rattana Leelawattana, Pat Korathanakhun.

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
