## [Decision Letter · Decision Letter 0]

21 Feb 2024

PONE-D-23-36251Factors associated with venous collaterals in patients with cerebral venous thrombosisPLOS ONE

Dear Dr. Korathanakhun,

Thank you for submitting your manuscript to PLOS ONE. After careful consideration, we feel that it has merit but does not fully meet PLOS ONE’s publication criteria as it currently stands. Therefore, we invite you to submit a revised version of the manuscript that addresses the points raised during the review process.

We look forward to receiving your revised manuscript.

Kind regards,

Suraj Shrestha, MBBS, DiMM

Academic Editor

PLOS ONE

Journal Requirements:

Additional Editor Comments:

Dear Authors,

Please find the reviewers comment and address these issues before we reconsider for publication.

Reviewer 1

The authors present a study on venous collaterals in VST. The paper is fairly novel and interesting study, although the paper is limited by small sample size, retrospective nature, not uniform approach to assessing collaterals. The discussion is well-written, although some portions such as the introduction need to be clearly written. I have several specific comments below:

In abstract, ensure that similar decimal places are used throughout (needs to be consistent)

Line 33 – the authors should be more specific, ie. In a cohort of patients with CVT, the frequency of collateral was …

Line 38 – would rephrase this, many would not consider VST a type of stroke.

Line 39 – this is redundant and out of place, please omit

Line 45 – should say that it can lead to hemorrhage.

Line 57 – the authors need to clearly state why this study is being done, despite so many prior studies on VST and collaterals. What is unique about this study? What are the knowledge gaps and how does this study seek to fulfill the gap?

Line 65 – why not CTV?

Line 85 – needs reference

Line 156 – please include frequency of patients with collaterals as well

- Why was malignancy not included as a risk factor?

- Was traumatic VST included? If not, the authors should clarify these are spontaneous

- Was is the interrater reliability of grading these collaterals?

- Not using gold standard uniform imaging in all patients should also be listed as limitation. For example, it likely easier to see collateral on DSA vs MRI.

Reviewer 2

Thank you very much for your submission to the PLOS ONE journal. This descriptive study sought to determine clinical and radiologic factors associated with the presence of venous collaterals in patients with cerebral venous thrombosis. I was impressed that you carried out a 20-year retrospective study to collect data for several different variables about 79 patients diagnosed with cerebral venous thrombosis (20 with venous collaterals and 59 without). Based on this, you determined that patients with venous collaterals were more likely to have CVT located in the superior sagittal sinus, and more likely to have had hormonal exposure. Overall, I found that your study was thorough, your methodology and data analyses sound, and your manuscript well-organized and clearly written. I believe that your manuscript could benefit from some revisions particularly to strengthen your discussion prior to resubmitting. Please find below some of my suggestions for your consideration.

Major comments

1. Your study describes some differences between patients with venous collaterals and those without, although you also mention how both the presentation and clinical outcomes in these two groups are not significantly different. May you elaborate on the clinical significance of the factors you identified as different between your two groups? How do you foresee the findings from your study being applied by neuroradiologists and neurologists evaluating patients with suspected or confirmed CVT?

2. You mention that the prevalence of venous collaterals in your population was much lower than that seen in previous studies, including a study that used the same classification system as your study (the Qureshi classification) but found a prevalence of 88% (versus 25% in your study). Are there specific differences between the characteristics of your populations that would be expected to account for such a major difference in prevalence, or how can we best reconcile these differences?

Thank you.

Reviewers' comments:

Reviewer's Responses to Questions

**Comments to the Author**

1. Is the manuscript technically sound, and do the data support the conclusions?

Reviewer #1: Yes

Reviewer #2: Yes

2. Has the statistical analysis been performed appropriately and rigorously? 

Reviewer #1: Yes

Reviewer #2: Yes

3. Have the authors made all data underlying the findings in their manuscript fully available?

Reviewer #1: Yes

Reviewer #2: Yes

4. Is the manuscript presented in an intelligible fashion and written in standard English?

Reviewer #1: Yes

Reviewer #2: Yes

5. Review Comments to the Author

Reviewer #1: The authors present a study on venous collaterals in VST. The paper is fairly novel and interesting study, although the paper is limited by small sample size, retrospective nature, not uniform approach to assessing collaterals. The discussion is well-written, although some portions such as the introduction need to be clearly written. I have several specific comments below:

In abstract, ensure that similar decimal places are used throughout (needs to be consistent)

Line 33 – the authors should be more specific, ie. In a cohort of patients with CVT, the frequency of collateral was …

Line 38 – would rephrase this, many would not consider VST a type of stroke.

Line 39 – this is redundant and out of place, please omit

Line 45 – should say that it can lead to hemorrhage.

Line 57 – the authors need to clearly state why this study is being done, despite so many prior studies on VST and collaterals. What is unique about this study? What are the knowledge gaps and how does this study seek to fulfill the gap?

Line 65 – why not CTV?

Line 85 – needs reference

Line 156 – please include frequency of patients with collaterals as well

- Why was malignancy not included as a risk factor?

- Was traumatic VST included? If not, the authors should clarify these are spontaneous

- Was is the interrater reliability of grading these collaterals?

- Not using gold standard uniform imaging in all patients should also be listed as limitation. For example, it likely easier to see collateral on DSA vs MRI

Reviewer #2: Thank you very much for your submission to the PLOS ONE journal. This descriptive study sought to determine clinical and radiologic factors associated with the presence of venous collaterals in patients with cerebral venous thrombosis. I was impressed that you carried out a 20-year retrospective study to collect data for several different variables about 79 patients diagnosed with cerebral venous thrombosis (20 with venous collaterals and 59 without). Based on this, you determined that patients with venous collaterals were more likely to have CVT located in the superior sagittal sinus, and more likely to have had hormonal exposure. Overall, I found that your study was thorough, your methodology and data analyses sound, and your manuscript well-organized and clearly written. I believe that your manuscript could benefit from some revisions particularly to strengthen your discussion prior to resubmitting. Please find below some of my suggestions for your consideration.

Major comments

1. Your study describes some differences between patients with venous collaterals and those without, although you also mention how both the presentation and clinical outcomes in these two groups are not significantly different. May you elaborate on the clinical significance of the factors you identified as different between your two groups? How do you foresee the findings from your study being applied by neuroradiologists and neurologists evaluating patients with suspected or confirmed CVT?

2. You mention that the prevalence of venous collaterals in your population was much lower than that seen in previous studies, including a study that used the same classification system as your study (the Qureshi classification) but found a prevalence of 88% (versus 25% in your study). Are there specific differences between the characteristics of your populations that would be expected to account for such a major difference in prevalence, or how can we best reconcile these differences?

Thank you again for your submission and I hope that my comments will be helpful to your group.

6. PLOS authors have the option to publish the peer review history of their article (what does this mean?). If published, this will include your full peer review and any attached files.

Reviewer #1: **Yes: **Alvin Das

Reviewer #2: No

---

## [Author Response · Author response to Decision Letter 0]

27 Feb 2024

Dear reviewers,

I would like to appreciate your sharp sight in this manuscript and valuable comments. Please find the response to reviewers’ comments in attached file.

Sincerely Yours, 

Pat Korathanakhun, M.D. 

---

## [Editor Report · Decision Letter 1]

28 Mar 2024

Factors associated with venous collaterals in patients with cerebral venous thrombosis

PONE-D-23-36251R1

Dear Dr. Korathanakhun,

We’re pleased to inform you that your manuscript has been judged scientifically suitable for publication and will be formally accepted for publication once it meets all outstanding technical requirements.

Kind regards,

Suraj Shrestha, MBBS, DiMM

Academic Editor

PLOS ONE
---

## [Editor Report · Acceptance letter]

3 Apr 2024

PONE-D-23-36251R1 

PLOS ONE

Dear Dr. Korathanakhun, 

I'm pleased to inform you that your manuscript has been deemed suitable for publication in PLOS ONE. Congratulations! Your manuscript is now being handed over to our production team.

Kind regards, 

on behalf of

Dr. Suraj Shrestha 

Academic Editor

PLOS ONE